# Periodic Solutions in a Simple Delay Differential Equation

Anatoli Ivanov [1,†] and Sergiy Shelyag [2,*,†]

[1]   Department of Mathematics, Pennsylvania State University, Dallas, PA 18612, USA; afi1@psu.edu
[2]   College of Science and Engineering, Flinders University, Tonsley, Adelaide, SA 5042, Australia
[*]   Correspondence: sergiy.shelyag@flinders.edu.au
[†]   These authors contributed equally to this work.

**Abstract:** A simple-form scalar differential equation with delay and nonlinear negative periodic feedback is considered. The existence of several types of slowly oscillating periodic solutions is shown with the same and double periods of the feedback coefficient. The periodic solutions are built explicitly in the case with piecewise constant nonlinearities involved. The periodic dynamics are shown to persist under small perturbations of the equation, which make it smooth. The theoretical results are verified through extensive numerical simulations.

**Keywords:** delay differential equations; periodic negative feedback; slowly oscillating solutions; periodic solutions; piecewise constant nonlinearities; explicit piecewise affine solutions; reduction to interval maps

## 1. Introduction

The simple-form delay differential equation (DDE),

$$x'(t) = -\mu x(t) + f(x(t - \tau)) \tag{1}$$

represents a broad variety of complex dynamical behaviors that can happen in delay differential models. It is also one of the most studied equations in the field, exhibiting a range of diverse dynamics from the global stability of equilibria to the instability and existence of periodic solutions and chaotic behaviours (in contrast to analogous ordinary differential equations). Basic and fundamental facts about the DDE (1) can be found in many sources; see, e.g., monographs [1–4], among others.

Equation (1) is also extensively used as a mathematical model for a variety of real world phenomena, most notably in biological applications; see, e.g., the monographs [1,4–6] and further references therein. Among the well-known and most studied models of type (1) are the Mackey–Glass physiological equations [5,7], the Wazewska–Lasota blood cell model [8], Nicholson's blowfly equations [9,10], and several others [1,4,6].

Some of the more accurate and adequate mathematical models are explicitly time dependent, which take into account certain intrinsic periodicity factors, such as, e.g., circadian rhythms and seasonal changes. Examples of such differential delay models can be found in publications [5,6,11–14]. Therefore, a natural generalization of Equation (1) is the following DDE:

$$x'(t) = -\mu(t)x(t) + a(t)f(x(t - \tau)), \tag{2}$$

where $\mu(t)$ and $a(t)$ are $T$-periodic functions, and the nonlinearity $f(x)$ satisfies appropriate feedback conditions. An important theoretical and applied question then arises: whether DDE (2) admits periodic solutions, which are induced by the periodicity in the coefficients $\mu$ and $a$. Therefore, the mathematical problem to address is to derive conditions under which Equation (2) possesses a nontrivial $T$-periodic solution. This research work is our initial attempt at answering this question for some specific partial cases of the equation under consideration.

This paper deals with the problem of the existence of periodic solutions for the simplest possible scalar differential delay equation of type (2) when $\mu(t) \equiv 0$:

$$x'(t) = a(t) f(x(t - \tau)), \tag{3}$$

where $f(x)$ is a continuous function satisfying the negative feedback assumption $x \cdot f(x) < 0$, $\forall x \neq 0$; the coefficient $a(t) \geq 0$ is a continuous periodic function with a period $T$ satisfying $T > \tau$; and $\tau > 0$ is a delay. The principal problem we are addressing in this work is to derive conditions on the parameters and nonlinearities of Equation (3), which would yield the existence of periodic solutions with the same period $T$ as the coefficient $a(t)$. Though there are some results on the existence of periodic solutions for the DDEs with periodic coefficients, none of them are applicable to the simplest form (Equation (3)), to the best of our assessment and knowledge. In the many cases that have been published in the research literature, generalization from the autonomous case to the periodic one leads to the loss of the steady positive equilibrium. In some other cases, the equilibrium state persists; however, the sufficient conditions for the existence of periodic solutions cannot distinguish between the nontrivial periodic solutions and the steady state. See more details on such relevant results and further references in, e.g., [11–15].

We are stating and dealing with the problem of the existence of nontrivial periodic solutions in simple-form DDE (3). Due to the negative feedback assumption on $f$, this equation always admits the trivial constant solution $x(t) \equiv 0$. The mathematical problem is to find periodic solutions that are distinct from the trivial one. This is a difficult problem, which has been largely not addressed on the general theoretical level. One way to pave a path for a systematic analysis in this direction is to consider model equations of the form given by Equation (3) with a simple $f$ and $a$.

We first construct the periodic solutions in an explicit form using piecewise constant functions (both the nonlinearity $f$ and the coefficient $a$). Then, we modify both functions to be continuous and "close" to the piecewise constant ones by "smoothing" them in a small neighborhood of the discontinuity set (they can also be of the $C^\infty$ type). The periodic dynamics and their stability are shown to persist under such small perturbations.

The rest of this paper is structured as follows. The second section "Preliminaries" contains the basics of DDE (3) and preliminary results from our recent conference papers [16,17], which are necessary for the discussion in subsequent sections. Section 3 deals with the periodic solutions of Equation (3) explicitly constructed for the initially piecewise constant nonlinearity $f$ and the coefficient $a$. The dynamics on a set of slowly oscillating solutions are reduced to those for induced one-dimensional maps, thus proving both the existence of periodic solutions and the types of their stability. In Section 3, it is shown that the dynamics of the interval maps and the periodic solutions from Section 2 persist when the defining functions $f$ and $a$ of Equation (3) are replaced with continuous or smoothed functions close to them. The arguments of the small regular perturbations of dynamics created by DDE (3) are used, which are behind such "small modifications". The theoretical derivations and exact analytical calculations are verified using extensive numerical simulations.

## 2. Preliminaries

In this work, we focus on the particular case $\mu = 0$ and $\tau = 1$ when Equation (1) becomes

$$x'(t) = a(t) f(x(t - 1)). \tag{4}$$

Note that the case of general delay $\tau > 0$ is easily reduced to the standard consideration $\tau = 1$ through the time scaling $t = \tau s$.

Given an initial function $\varphi(s) \in C([-1,0], \mathbb{R}) := \mathbb{X}$ in the standard phase space $\mathbb{X}$, the corresponding solution $x = x(t, \varphi)$ to Equation (4) is easily found for all $t \geq 0$ through consecutive forward integration (the step method). Let $S^t(\varphi)$ be the shift in $\mathbb{X}$ by time $t$ along the solutions of delay differential Equation (4), that is, $S^t(\varphi) = x(t + s, \varphi), s \in [-1, 0]$.

It is a straightforward observation that a fixed point $\varphi_0$ of the shift by period $T$ ($S^T(\varphi_0) = \varphi_0$) gives rise to a periodic solution $x(t, \varphi_0)$ of DDE (4). Under such an interpretation, the zero solution $x(t) \equiv 0$ is a result of the trivial fixed point $\varphi_0(s) \equiv 0, s \in [-1, 0]$, which always exists for DDE (4), due to the negative feedback assumption. The mathematical problem is to derive conditions for the existence of nontrivial fixed points of the shift operator $S^T$, which is a principal objective of this work.

A solution is called oscillating if it has an infinite sequence of zeros $\{t_n\} \to \infty, x(t_n) = 0$, $n \in \mathbb{N}$, and $x(t)$ is not an identical zero eventually (on any interval of the form $[t_0, \infty)$). An oscillating solution is called slowly oscillating if the sequence of zeros is such that the distance between consecutive zeros is greater than the delay, i.e., $t_{n+1} - t_n > 1, \forall n \in \mathbb{N}$.

Sufficient conditions for the oscillation of all solutions to Equation (4) are well known; see, e.g., Ref. [18] and further references therein. With the negative feedback assumption $x f(x) < 0, x \neq 0$, and the non-negativity of the coefficient $a(t) \geq 0, a(t) \not\equiv 0$, either $f'(0)$ is sufficiently large, $|f'(0)| \geq f_1 > 0$, and the coefficient $a$ is separated from zero, $a(t) \geq a_1 > 0$, or $f'(0) \neq 0$, and $a_1 > 0$ is large enough (in both cases, the product $|f_1 \cdot a_1|$ must be sufficiently large). These conditions are satisfied for our considerations to follow in the following sections, where the functions $f$ and $a$ are smoothed continuous ones derived from the initial piecewise constant functions (the piecewise constant case can be viewed as a limit of the continuous one).

In this paper, we deal with the slowly oscillating solutions to Equation (4). The slowly oscillating solutions are obtained when the initial functions belong to one of the two sign-definite cones. Consider the following two standard subsets of initial functions:

$$K_- := \{\varphi \in \mathbb{X} \mid \varphi(s) < 0 \ \forall s \in [-1, 0]\} \quad \text{and} \quad K_+ := \{\varphi \in \mathbb{X} \mid \varphi(s) > 0 \ \forall s \in [-1, 0]\}.$$

Assuming that all solutions to Equation (4) oscillate, it is a straightforward calculation to verify that for arbitrary $\varphi \in K_-$, the corresponding solution $x(t; \varphi)$ has an increasing sequence of zeros $0 < t_1 < t_2 < t_3 < \cdots$ such that $t_{k+1} - t_k > \tau = 1, k \in \mathbb{N}$, and

$$x(t) > 0 \ \forall t \in (t_{2k-1}, t_{2k}) \quad \text{and} \quad x(t) < 0 \ \forall t \in (t_{2k}, t_{2k+1}). \tag{5}$$

In the case when $\varphi \in K_+, \varphi(0) = h > 0$, the above inequalities on the consecutive intervals $(t_{2k-1}, t_{2k})$ and $(t_{2k-1}, t_{2k})$ are reversed into the opposite. Therefore, the solution $x(t; \varphi)$ is slowly oscillating. The consecutive intervals $(t_j, t_{j+1}), j \in \mathbb{N}$, are called semicycles.

Based on well-known facts [18] and the reasoning above, we have the following statement.

**Proposition 1.** *Suppose that the nonlinearity $f \in C(\mathbb{R}, \mathbb{R})$ satisfies the negative feedback assumption $x \cdot f(x) < 0, \forall x \neq 0$, and $f'(0) < 0$. Let the coefficient $0 < a(t) \in C([0, \infty), \mathbb{R})$ be T-periodic such that $\min\{a(t), t \in [0, T]\} \geq m_0 > 0$. Assume that $|f'(0)| \cdot m_0 > 1/e$. Then, for every initial function $\varphi \in \mathbb{K}_-$ with $\varphi(0) = h < 0$, the corresponding solution $x(t, h), t \geq 0$, is slowly oscillating with the sequence of simple zeros $\{t_n\}$ satisfying the inequalities (5) on the consecutive semicycles. Likewise, for every initial function $\psi \in K_+$ with $\psi(0) = h > 0$, the corresponding solution $x(t, \psi), t \geq 0$, is slowly oscillating, satisfying the inequalities opposite to those in (5).*

Additional related theoretical basics on delay differential equations can be found in, e.g., monographs [2,3].

## 3. Periodic Solutions

In this section, the periodic solutions of interest are explicitly constructed based on model piecewise constant nonlinearity $f$ and the periodic coefficient $a$. Let $f(x) = f_0(x) = -\text{sign}(x)$ and define $a = a(t, p_1, p_2, a_1, a_2) := A_0(t)$ as follows:

$$a(t) = A_0(t) = \begin{cases} a_1 > 0, & \text{if } 0 \leq t < p_1 \\ a_2 > 0, & \text{if } p_1 \leq t < p_1 + p_2 \\ \text{periodic extension outside } [0, p_1 + p_2) \text{ for all } t \in \mathbb{R}. \end{cases} \tag{6}$$

Let an initial function $\varphi(s) \in \mathbb{X}$ be such that either $\varphi(s) < 0$ or $\varphi(s) > 0, \forall s \in [-1, 0]$, and $\varphi(0) = h \neq 0$. Clearly, the corresponding solution $x = x(t, \varphi)$ depends on the value $h$ only and does not depend on the particular values of $\varphi(s) \neq 0$ for $s \in [-1, 0)$. In addition, it is uniquely determined by the parameters $p_1, p_2, a_1,$ and $a_2$ of the function $a(t)$. The solution $x$ is slowly oscillating and piecewise linear (affine) for $t \geq 0$.

Like in the case of continuous $f$ and $a$, the corresponding solutions to DDE (4) with $f = f_0$ and $a = a_0$ are slowly oscillating with the consecutive zeros $\{t_n\}$ satisfying the inequalities (5). In this case, we do not need any additional assumptions on $f_0$ or $a_0$ as in Proposition 1, as the lower rates of growth or decay of any such solution are bounded away from zero (the rates of growth/decay are also bounded from above, due to the boundedness of $f$ and $a$). However, since the constructed solutions are slowly oscillating, we assume that, necessarily, the assumption $p_1 + p_2 > 1$ must be in place.

In this section, we follow and expand on ideas and explicit constructions from our recent work [16,17]. In particular, functions $f$ and $a$ are of the same form as in these papers.

### 3.1. Periodic Solutions with Coefficient Period (Type I)

In this subsection, we explicitly construct piecewise affine periodic solutions to Equation (4), which are of the same period $T > 1$ as the periodic coefficient $a(t)$. Some of the exposition elements are close and based on results in our recent conference paper [17].

We presume that the desired periodic solutions have the shape as shown in Figure 1. That is, the initial value $\varphi(0) = h < 0$ is determined by an initial function $\varphi(s) \in K_-$. The corresponding solution $x = x(t, \varphi), t \geq 0$, has exactly two zeros $\{t_1, t_2\}$ on the initial period $[0, T], T = p_1 + p_2$, with $t_1 < t_2 = t_1 + 2 < T$, for particular assumptions on the parameters $p_1, p_2, a_1,$ and $a_2$. Such assumptions were verified to be valid through the analytical calculations that follow, and they were confirmed using corresponding numerical simulations.

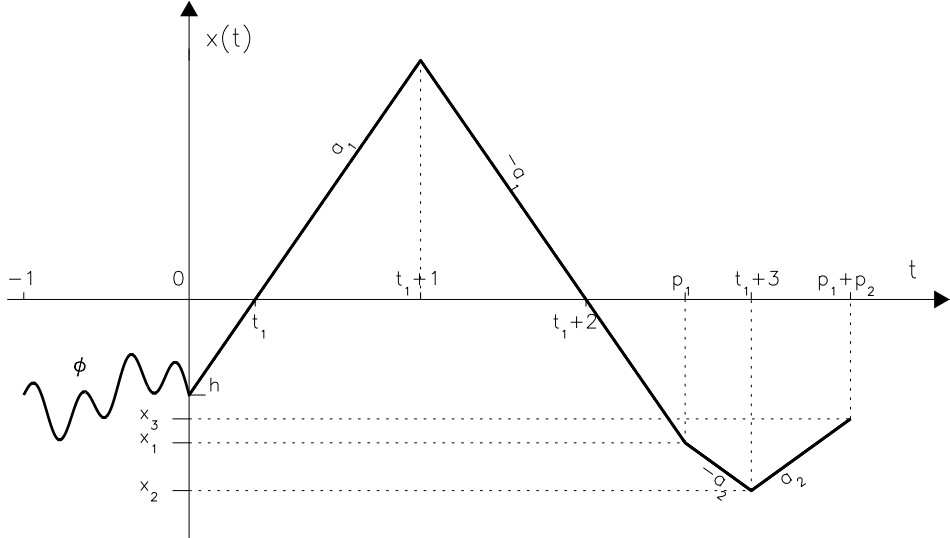

**Figure 1.** Slowly oscillating piecewise affine solution (Type I).

It is straightforward to find the desired values of the solution $x = x(t, h)$ for any $t \in [0, T]$. We have the following calculations:

$$t_1 = -\frac{h}{a_1}, \quad x_1 = x(p_1) = -h + 2a_1 - a_1 p_1,$$

$$x_2 = x(t_1 + 3) = \left(\frac{a_2}{a_1} - 1\right)h + a_1(2 - p_1) + a_2(p_1 - 3),$$

$$x_3 = x(T) = x(p_1 + p_2) = \left(2\frac{a_2}{a_1} - 1\right)h + a_1(2 - p_1) + a_2[(2p_1 + p_2) - 6] := G(h),$$

where the coefficients of the affine function $G(h) := mh - b$ are given by

$$m = 2\frac{a_2}{a_1} - 1, \quad \text{and} \quad b = a_1(p_1 - 2) + a_2[6 - (2p_1 + p_2)]. \tag{7}$$

It is elementary to deduce that the range for the slope $m$ is the open interval $(-1, \infty)$ with $|m| < 1$ when $0 < a_2 < a_1$ holds, and $m > 1$ when $0 < a_1 < a_2$ is valid.

A fixed point $h_* < 0$ of the map $G(h) = mh - b$ gives rise to a slowly oscillating periodic solution $x = x(t, h_*)$ for Equation (4). Moreover, by the affine type of $G$ and the construction, such a periodic solution is asymptotically stable if $|m| < 1$, which is equivalent to $0 < a_2 < a_1$. The periodic solution is unstable when $m > 1$, which is equivalent to $0 < a_1 < a_2$. The unique fixed point $x = h_*$ is easily found as follows:

$$h_* = b/(m-1) = \frac{a_1[a_1(p_1 - 2) + a_2[6 - (2p_1 + p_2)]]}{2(a_2 - a_1)}. \tag{8}$$

Therefore, we arrive at the following statement.

**Theorem 1.** *Let the parameters $a_1, a_2, p_1$, and $p_2$ be given with the values $m$ and $b$ defined by (7). Then, consider the following:*

(a) *In a case when $|m| < 1$ $(0 < a_2 < a_1)$ and $b > 0$ are valid, DDE (4) has an asymptotically stable slowly oscillating T-periodic solution.*

(b) *In the complimentary case when $m > 1$ $(0 < a_2 < a_1)$ and $b < 0$ are valid, the equation has an unstable slowly oscillating T-periodic solution.*

*The periodic solutions are generated by the initial function $\varphi(s) \equiv h_*$, $s \in [-1, 0]$, where $h_*$ is given by (8).*

**Remark 1.** (a) *The local stability case $|m| < 1$ is equivalent to the relationship $a_1 > a_2 \geq 0$. At the same time, in order for the fixed point to be negative, one must assume that $b = a_1(p_1 - 2) + a_2[6 - (2p_1 + p_2)] > 0$. The inequality $p_1 > 2$ is valid based on the construction. For arbitrary fixed $p_1 > 2$, $p_2 > 0$, and $a_2 > 0$, there exists $a_1^0$ such that the inequalities $|m| < 1$ and $b > 0$ are satisfied for all $a_1 \geq a_1^0$. Thus, the required assumptions are met for all sufficiently large $a_1$. This case is treated in more detail in our recent paper [17].*

(b) *The instability case $m > 1$ is equivalent to $0 < a_1 < a_2$ (note that the case $m < -1$ is not possible for any choices of positive $a_1$ and $a_2$). In order to have the fixed point $h_*$ to be negative, one must assume that $b < 0$ is satisfied, that is $a_1(p_1 - 2) + a_2[6 - (2p_1 + p_2)] < 0$. Since $p_1 > 2$, the assumption $p_2 \geq 2$ would imply that $6 - (2p_1 + p_2) < 0$. Therefore, given fixed values of $a_1 > 0$, $p_1 > 2$, and $p_2 \geq 2$, there exists $a_2^0$ such that $b < 0$ for all $a_2 \geq a_2^0$. Thus, the required assumptions are met for all sufficiently large $a_2$.*

(c) *It is clear that due to the continuous dependence of $m$ and $b$ on the parameters $a_1, a_2, p_1$, and $p_2$, the stable or unstable periodic solutions of Theorem 1 exist on an open box in $\mathbb{R}^4$, each side of which is a corresponding open interval about a specific value of each of the four parameters corresponding to a fixed point $h_*$. These boxes can be multiple, as our numerical insight indicates, and they do not overlap for the stable and unstable solutions due to the respective opposite relationship between $a_1$ and $a_2$.*

The parametric range for which Theorem 1 guarantees the existence of slowly oscillating periodic solutions, either stable or unstable, is large.

Note that due to the symmetry property of the nonlinearity $f$ (oddness) and the procedure of the construction of the periodic solutions $x_{h_*}$, under the assumption of Theorem 1, there also exists a periodic solution symmetric to $x_{h_*}$ generated by the initial function $\phi(s) \equiv -h_* > 0$. The two periodic solutions are related by $x_{-h_*}(t) \equiv -x_{h_*}(t)$; thus, they are symmetric to each other.

A large sample of parametric values has been derived for which numerical periodic solutions have been obtained and confirmed to be of the described form. Table 1 is a sample selection of parametric values resulting in asymptotically stable slowly oscillating periodic solutions given by Theorem 1. The values $h_* < 0$ for these periodic solutions are easily found from the explicit formula, Equation (8). They have also been verified to exist numerically.

**Table 1.** Examples of parametric values for which stable $T$-periodic solutions of Equation (4) with coefficient $a$ defined by (6) have been demonstrated numerically.

| $a_1$ | $a_2$ | $p_1$ | $p_2$ | $h_*$ | $T$ |
|---|---|---|---|---|---|
| 1 | 0.25 | 2.5 | 1.5 | $-0.25$ | 4 |
| 2 | 0.5 | 2.5 | 2 | $-1/3$ | 4.5 |
| 2 | 0.25 | 2.5 | 1 | $-4/7$ | 3.5 |
| 1 | 0.5 | 3 | 1 | $-0.5$ | 4 |
| 2 | 1 | 3 | 1.5 | $-0.5$ | 4.5 |
| 2.5 | 0.5 | 3 | 4 | $-0.31$ | 7 |
| 3 | 0.5 | 3 | 4.5 | $-0.45$ | 7.5 |
| 4 | 1 | 3.5 | 2 | $-2$ | 5.5 |
| 5 | 0.5 | 3 | 3 | $-1.94$ | 6 |
| 5 | 1 | 3 | 2 | $-1.88$ | 5 |
| $\sqrt{10}$ | $1/\sqrt{5}$ | $\pi$ | $e+1$ | $-1.0602$ | $\pi+e+1$ |

The next table, Table 2, is a sample selection of unstable slowly oscillating periodic solutions implied by Theorem 1. The values $h_*$ for these periodic solutions are found from Equation (8) as well. The unstable periodic solutions cannot be verified numerically as respective initial functions $\varphi \equiv h_* < 0$ actually generate approximately close, nearby solutions (not the exact ones). These solutions, when calculated in forward time, are repelled for large times by the actual unstable ones and are attracted by the stable periodic solutions of the double period $2T$. The latter are studied analytically and numerically and described in detail in the next subsection.

**Table 2.** Examples of parametric values for which unstable periodic solutions exist for Equation (4) with the coefficient defined by (6). The values $h_*$ are given by (8) with $b < 0$ and $m > 1$.

| $a_1$ | $a_2$ | $p_1$ | $p_2$ | $h_*$ | $T$ |
|---|---|---|---|---|---|
| 0.5 | 5 | 5 | 1 | $-1.31$ | 6 |
| 1 | 5 | 4 | 1 | $-1.625$ | 5 |
| 1 | 7 | 4 | 1 | $-1.5$ | 5 |
| 1 | 7 | 3 | 2 | $-1.0833$ | 5 |
| 1.5 | 7 | 2.5 | 2.5 | $-1.3295$ | 5 |
| 1.5 | 7 | 3.5 | 1.5 | $-2.0795$ | 5 |
| 1.5 | 7 | 4 | 3 | $-4.3636$ | 7 |
| 2 | 7 | 3 | 2 | $-2.4$ | 5 |
| 2 | 7 | 3 | 3 | $-4.6$ | 6 |
| 2 | 7 | 3.5 | 2.5 | $-4.3$ | 6 |
| 2.5 | 7 | 2.5 | 2.5 | $-1.2614$ | 5 |
| 2.5 | 7 | 3 | 2 | $-1.5682$ | 5 |
| $1/\sqrt{5}$ | $3\pi/2$ | $2e$ | $\pi/3$ | $-1.382$ | $2e+\pi/3$ |

### 3.2. Periodic Solutions with Double Coefficient Period (Type II)

In this subsection, we explicitly construct piecewise affine periodic solutions to Equation (4), which are of the double period $2T$ of the $T$-periodic coefficient $a(t)$. Some of the exposition is close and based on the results of our recent conference paper [16].

We presume that the desired periodic solutions have the initial shape shown in Figure 2. That is, the initial value $\varphi(0) = h < 0$ is determined using an initial function $\varphi(s) \in K_-$. The corresponding solution $x = x(t, \varphi), t \geq 0$, has exactly one zero $t_1$ on the initial periodic interval $[0, T], T = p_1 + p_2$, with $t_1 < p_1$ and $x(t) < 0 \; \forall t \in [0, t_1)$ and $x(t) > 0 \; \forall t \in (t_1, T]$. Such assumptions are verified to be valid through the analytical calculations that follow, as well as confirmed using associated numerical simulations.

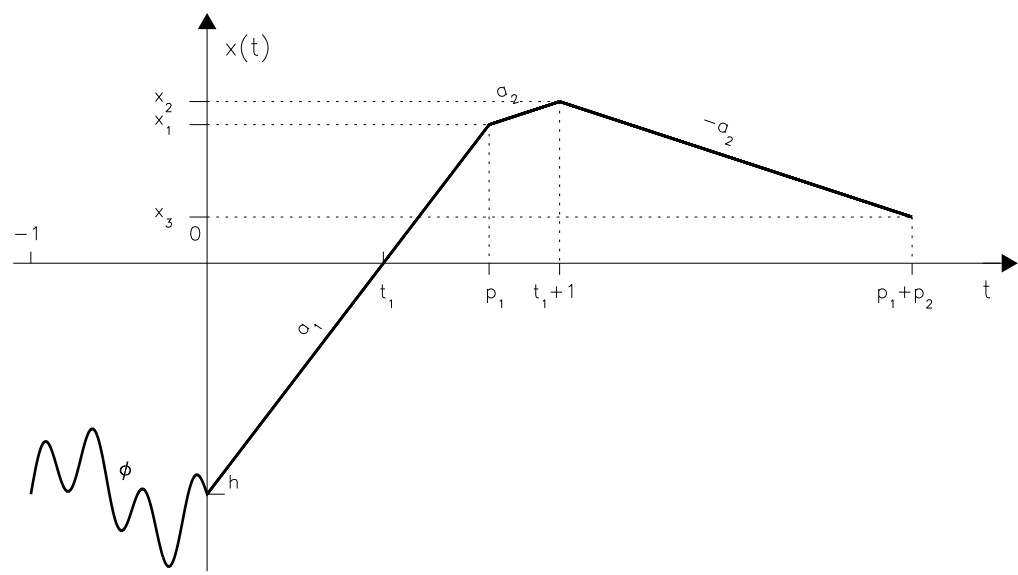

**Figure 2.** Slowly oscillating piecewise affine solution (Type II) on initial periodic interval.

It is straightforward to calculate the desired values of $t_1, x_1, x_2, x_3$ as follows:

$$t_1 = -\frac{h}{a_1}, \quad x_1 = x(p_1) = h + a_1 p_1, \quad x_2 = x(t_1 + 1) = \left(1 - \frac{a_2}{a_1}\right) h + p_1(a_1 - a_2) + a_2,$$

$$x_3 = x(p_1 + p_2) = \left(1 - 2\frac{a_2}{a_1}\right) h + a_1 p_1 + a_2(2 - 2p_1 - p_2) := F_1(h) = kh + d,$$

where

$$k = 1 - 2\frac{a_2}{a_1} \quad \text{and} \quad d = a_1 p_1 + a_2(2 - 2p_1 - p_2) \tag{9}$$

are uniquely defined by the parameters $p_1, p_2, a_1,$ and $a_2$.

Likewise, when analogous calculations $\psi \in K_+$ and $\psi(0) = h > 0$ yield

$$x_3 = x(p_1 + p_2, \psi) = \left(1 - 2\frac{a_2}{a_1}\right) h - a_1 p_1 - a_2(2 - 2p_1 - p_2) := F_2(h) = kh - d,$$

with the same $k$ and $d$ as in the expression for $F_1$ given by Equation (9), it is easy to see that for arbitrary positive $a_1$ and $a_2$, the respective values of $k$ can only belong to the range $(-\infty, 1)$ with $|k| < 1$ if $0 < a_2 < a_1$ and $k < -1$ if $0 < a_1 < a_2$.

We are interested in parametric values $p_1, p_2, a_1$ and $a_2$ and initial values $h < 0$ such that $h_1 = F_1(h) > 0$ and $h_2 = F_2(h_1) < 0$. This will be the case when $d > 0$, at least in some vicinity $|h| < \delta_1$, where $h \neq 0$, of the discontinuity point $h = 0$. The through map $F_0 = F_2 \circ F_1$ is given by $F_0(h) = k^2 h + (k - 1)d$. Its only fixed point is found as $h_* = -d/(k + 1)$, and it is negative when $d > 0$ and $|k| < 1$. The fixed point $h_*$ generates an asymptotically stable periodic solution of period $2T$ when the initial function for Equation (4) is chosen as $\varphi(s) \equiv h_*$. The periodic solution can also be described in terms

of a cycle of period 2 for an appropriate one-dimensional map $F$ defined by the maps $F_1$ and $F_2$ introduced above. Namely, we define the piecewise affine map $F$ for $h \in \mathbb{R} \setminus \{0\}$ as follows:

$$F(h) = \begin{cases} F_1(h) = kh + d, & \text{if } h < 0 \\ F_2(h) = kh - d, & \text{if } h > 0. \end{cases}$$

The map $F$ is discontinuous at $h = 0$ and is symmetric (odd) with respect to $h = 0$, $F(-h) = -F(h)$, $\forall h \neq 0$. Assuming $d > 0$, one can see that the negative feedback condition is satisfied at least locally in a vicinity of $h = 0$. The global dynamics of map $F$ are very simple in this case, as described by the following statement.

**Proposition 2.** *Suppose that $d > 0$. Then, consider the following:*

(i) *If $0 < a_2 < a_1$ is satisfied, map $F$ has a unique globally attracting cycle of period 2, given by $\{h_*, -h_*\}$, where $h_* = -d/(k+1)$. The two-cycle attracts all the initial values $h \in \mathbb{R} \setminus \{0\}$ in the case when $a_1 > 2a_2$ holds, and all the initial values from the interval $h \in (-d/k, d/k)$ in the case when $2a_2 > a_1 > a_2$ holds;*

(ii) *If $0 < a_1 < a_2$ is satisfied, map $F$ has no cycles of period 2; moreover, any iterative sequence $F^n(h), h \neq 0$, diverges, $\lim_{n \to \infty} |F^n(h)| = \infty$.*

Therefore, by translating the properties of the interval map $F$ to the slowly oscillating solutions of DDE (4), we have the following statement.

**Theorem 2.** *Let the parameters $a_1, a_2, p_1$, and $p_2$ be given with values $k$ and $d$ defined by (9) and $d > 0$. Then, in the case when $|k| < 1$ $(0 < a_2 < a_1)$ is valid, DDE (4) has an asymptotically stable slowly oscillating $2T$-periodic solution. The periodic solution is generated by the initial function $\varphi(s) \equiv h_*, s \in [-1, 0]$, where $h_* = -d/(k+1)$. In the case when $k < -1$ $(0 < a_2 < a_1)$ is valid, the equation does not have any $2T$-periodic slowly oscillating solutions.*

Table 3 provides sample selections of parametric values for which the stable $2T$-periodic solutions exist. Besides using Formula (8) to calculate respective $h_*$, their existence was also confirmed numerically.

**Table 3.** Examplesof parametric values for which stable periodic solutions with the double period of coefficient $a(t)$ exists for Equation (4). The values of $h_*$ are given by Equation (9).

| $a_1$ | $a_2$ | $p_1$ | $p_2$ | $h_*$ | $2T$ |
|---|---|---|---|---|---|
| 4 | 1 | 0.5 | 2.5 | $-0.33$ | 6 |
| 4 | 1 | 1 | 3.5 | $-0.33$ | 9 |
| 5 | 1 | 1 | 3.5 | $-0.938$ | 9 |
| 5 | 1 | 0.5 | 3 | $-0.313$ | 7 |
| 6 | 1 | 1 | 3.5 | $-1.5$ | 9 |
| 6 | 1 | 1 | 4.5 | $-0.9$ | 11 |
| 6 | 1.5 | 1.0 | 3.5 | $-0.5$ | 9 |
| 7 | 1.5 | 1.5 | 3.5 | $-2.39$ | 10 |
| 7 | 2.5 | 2.5 | 2.5 | $-2.92$ | 10 |
| 7 | 2.5 | 2 | 3 | $-1.17$ | 10 |
| $1/\sqrt{5}$ | $3\pi/2$ | $3\pi/2$ | $e/2$ | $-1.12$ | $3\pi + e$ |

### 3.3. Coexistence of Periodic Solutions

Any slowly oscillating solution considered above in Sections 3.1 and 3.2 is uniquely defined by the sign-definite initial function/values $\varphi \in \mathbb{X}$ such that $\varphi(0) = h \neq 0$ and $\varphi(s) \neq 0 \; \forall s \in [-1, 0]$, and by the parameters $p_1, p_2, a_1$, and $a_2$ of the periodic coefficient $a(t)$ (note that $f = f_0(x) = -\text{sign}(x)$ is fixed). One solves the initial value problem for $t \geq 0$ as follows:

$$x'(t) = a_0(t) f_0(x(t-1)), \; t \geq 0, \; \varphi \in \mathbb{X}, \varphi(0) = h \neq 0, \varphi(s) \neq 0 \; \forall s \in [-1, 0]. \tag{10}$$

Its solution $x_\varphi(t) = x_\varphi(t, h, p_1, p_2, a_1, a_2)$ is easily found to exist for all $t \geq 0$; it is uniquely determined by the set of the parameters $p_1, p_2, a_1$, and $a_2$ and the initial value of $h$ at $t_0 = 0$.

Instead of the initial time value $t_0 = 0$, as in the case above, one can consider a similar initial value problem for any $t_0 > 0$. In particular, in the case of the initial time $t_0 = p_1$, one is solving the following initial value problem for $t \geq p_1$:

$$x'(t) = a_0(t)\, f_0(x(t-1)),\ t \geq p_1,\ \psi(p_1) = h \neq 0,\ \psi(s) \neq 0\ \forall s \in [p_1 - 1, p_1]. \tag{11}$$

Similarly, its solution $x_\psi(t) = x_\psi(t, h, p_1, p_2, a_1, a_2)$ is uniquely defined by the set of parameters $p_1, p_2, a_1$, and $a_2$ and by the initial value $h$ at $t_0 = p_1$.

Now, due to the $T$-periodicity of the coefficient $a(t)$, one can easily see that the solutions of the two initial value problems, (10) and (11), are closely related, with an interchange in the set of the parameters and a time shift. Namely, there is just a time shift by $p_1$ with the parameter interchange in the following sense:

$$x_\varphi(t, h, p_2, p_1, a_2, a_1) = x_\psi(t + p_1, h_1, p_1, p_2, a_1, a_2), \quad \forall t \geq 0, \tag{12}$$

with an appropriate $h_1 = h_1(h)$. This simple observation allows us to conclude the simultaneous existence of periodic solutions of both types I and II, which are described in Theorems 1 and 2. Since the value of the ratio $a_1/a_2$ with respect to 1 determines the stability of these periodic solutions, their stability types must be opposite to each other. We have the following statement.

**Theorem 3.** *There is an open set of parameters $p_1, p_2, a_1$, and $a_2$ such that DDE (4) has simultaneously an unstable slowly oscillating $T$-periodic solution of type I and an asymptotically stable $2T$-periodic solution of type II. These two periodic solutions can be paired in the following sense: small perturbations of the unstable solution are repelled by it and attracted by a respective stable solution as $t \to \infty$.*

Such a situation of two coexisting periodic solutions is depicted in Figure 3. The unstable periodic solution $x_u(t)$ of period $T = 4$ is given by Theorem 1 when $p_1 = 3, p_2 = 1, a_1 = 1$, and $a_2 = 6$; the respective value of $h$ is $h_* = -0.5$. The stable periodic solution $x_s(t)$ of period $2T = 8$ is given by Theorem 2 for the parameter values $p_1 = 1, p_2 = 3, a_1 = 6$, and $a_2 = 1$ so that the respective values of $p_1, p_2$ and $a_1, a_2$ are interchanged for these two solutions. In addition, the solution $x_s(t)$ is shifted by $p_1$ to the right (graph (b) in Figure 3 shows the solution $x_s(t + p_1)$ for longer time intervals).

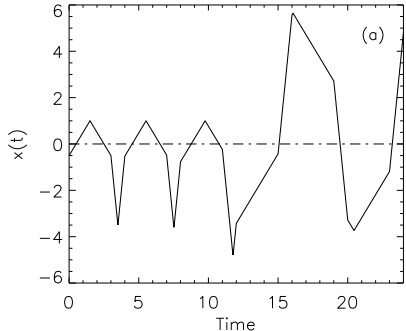 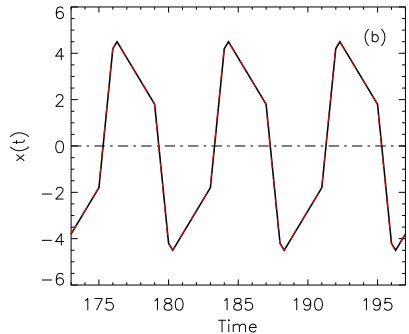

**Figure 3.** An example of numerical solution with $a_1 = 1, a_2 = 6, p_1 = 3$, and $p_2 = 1$. (**a**)—unstable solution with single period, which is destroyed after two first periods. (**b**)—stable solution with double period. Numerical solution of Equation (4) with $a_1 = 6, a_2 = 1, p_1 = 1$, and $p_2 = 3$, of which its solution has the double period of the $T$-periodic coefficient and is overplotted as a red dashed line in the right panel.

Tables 4 and 5 provide a sample selection of such dual cases of the coexistence of two types of stable/unstable solutions. Table 5 is a mirror image of Table 4 for the stable periodic solutions with double periods. Namely, each row in Table 5 represents a stable periodic solution of the corresponding unstable solution in the same row number as Table 4.

**Table 4.** Examples of parametric values for which unstable and stable periodic solutions coexist for DDE (4) with the coefficient defined by (6). The parametric data are shown for the unstable solutions.

| $a_1$ | $a_2$ | $p_1$ | $p_2$ | $h_*$ | $T$ |
|-------|-------|-------|-------|-------|-----|
| 0.5 | 2.5 | 3 | 0.5 | −0.09 | 3.5 |
| 0.5 | 3 | 5 | 1 | −1.35 | 6 |
| 0.5 | 5 | 4 | 0.5 | −0.64 | 4.5 |
| 0.5 | 7 | 3 | 2 | −0.52 | 5 |
| 1 | 6 | 3 | 1 | −0.5 | 4 |
| 1 | 7 | 5 | 1 | −2.67 | 6 |
| 2 | 7 | 2 | 3 | −1.4 | 5 |
| 3 | 7 | 4 | 1 | −5.62 | 5 |
| $\pi/6$ | $e$ | $\pi$ | $1/2$ | −0.18 | $\pi + 1/2$ |

**Table 5.** Examples of parametric values for which stable periodic solutions exist for Equation (4) with the coefficient defined by Equation (6). The parametric data are shown for the stable solutions.

| $a_1$ | $a_2$ | $p_1$ | $p_2$ | $h_*$ | $2T$ |
|-------|-------|-------|-------|-------|------|
| 2.5 | 0.5 | 0.5 | 3 | −0.156 | 7 |
| 3 | 0.5 | 1 | 5 | −0.3 | 12 |
| 5 | 0.5 | 0.5 | 4 | −0.56 | 9 |
| 7 | 0.5 | 2 | 3 | −6.19 | 10 |
| 6 | 1 | 1 | 3 | −1.8 | 8 |
| 7 | 1 | 1 | 5 | −1.17 | 12 |
| 7 | 2 | 3 | 2 | −6.3 | 10 |
| 7 | 3 | 1 | 4 | −4.375 | 10 |
| $e$ | $\pi/6$ | $1/2$ | $\pi$ | −1.92 | $2\pi + 1$ |

## 4. Smoothed Nonlinearities

In this section, we follow a standard procedure of replacing the discontinuous piecewise constant functions $f_0$ and $A_0$ with continuous functions $f_\delta$ and $A_\delta$ close to them for a small $\delta > 0$. This procedure has been used in numerous cases; typical examples can be found in papers [19–22]. In a $\delta$-neighborhood with every discontinuity point for both $f_0$ and $A_0$, the jump discontinuity is replaced with an affine function by connecting the two constant levels using a line segment. That is, define $f_\delta(x)$ and $A_\delta(t)$ as follows:

$$f(x) = f_\delta(x) = \begin{cases} +1 \text{ if } x \leq -\delta \\ -1 \text{ if } x \geq \delta \\ -(1/\delta)x \text{ if } x \in (-\delta, \delta), \end{cases} \tag{13}$$

and

$$a(t) = A_\delta(t) = \begin{cases} a_2 + \frac{a_1 - a_2}{2\delta}(t + \delta) \text{ if } t \in (-\delta, \delta) \\ a_1 \text{ if } t \in [\delta, p_1 - \delta] \\ a_1 + \frac{a_2 - a_1}{2\delta}[t - (p_1 - \delta)] \text{ if } t \in (p_1 - \delta, p_1 + \delta) \\ a_2 \text{ if } t \in [p_1 + \delta, p_1 + p_2 - \delta] \\ a_2 + \frac{a_1 - a_2}{2\delta}[t - (p_2 - \delta)] \text{ if } t \in (p_1 + p_2 - \delta, p_1 + p_2 + \delta) \\ \text{periodic extension on } \mathbb{R} \text{ outside interval } [0, p_1 + p_2). \end{cases} \tag{14}$$

We shall show next that the values of the piecewise affine slowly oscillating solutions constructed in Section 3 for $t \geq 0$ do not change throughout the entire periodic interval $[0, T]$, except in a small $\varepsilon$-vicinity of the corner points ($\varepsilon$ is a multiple of $\delta$).

Indeed, the case of a symmetric corner solution, as it passes through a $\delta$-neighborhood of $x = 0$ for the nonlinearity $f_\delta$, is shown in Figure 4 (which corresponds to, e.g., the type I solution of the first positive semicycle depicted in Figure 1). The $\delta$-neighborhood of $x = 0$ for $f$ corresponds to the $\varepsilon = \delta / a_1$ neighborhood for $t$ at $t = t_1 + 1$. It is immediately clear that the replacement of the corner type solution on the interval $[t_1 + 1 - \varepsilon, t_1 + 1 + \varepsilon]$ with the respective parabolic segment, as a result of the corresponding integration now with a linear nonlinearity $f$, results in a symmetric parabola with the vertex at $t_1 + 1$ and the same equal values $x(t_1 + 1 - \varepsilon)$ and $x(t_1 + 1 - \varepsilon)$ as for the initial piecewise solution with the discontinuous $f_0$.

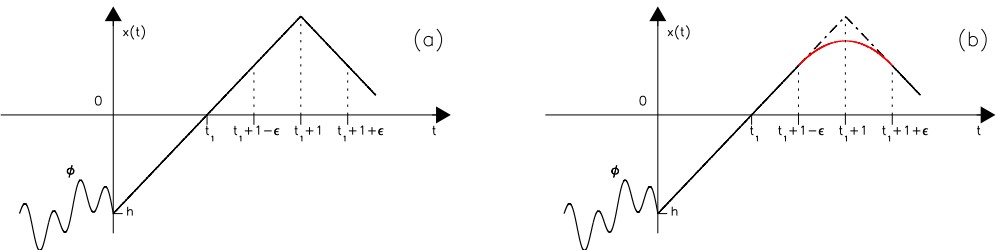

**Figure 4.** Sketches of (**a**) continuous solution $x$ with corner-type discontinuity for $x'$ at $t_1 + 1$ for functions $A_0(t)$ and $f_0(x)$; (**b**) $C^1$-smooth solution with parabolic matching on $[t_1 + 1 - \varepsilon, t_1 + 1 + \varepsilon]$, $\varepsilon a_1 = \delta$, for functions $A_\delta(t)$ and $f_\delta(x)$ with small $\delta > 0$.

The general case of a corner-type solution (piecewise affine) is shown in Figure 5, which corresponds to, e.g., point $(p_1, x_1)$ on the graph of a solution of type II (see Figure 2). We shall show now that a smooth parabolic segment solution connecting points $(p_1 - \varepsilon, x(p_1 - \varepsilon))$ and $(p_1 + \varepsilon, x(p_1 + \varepsilon))$ is the precise replacement of the corresponding piecewise affine segment solution that is obtained for the piecewise constant functions $f_0$ and $A_0$. The values of the solution $x$ and its derivatives for the piecewise affine solution are given as follows:

$$x(p_1 - \varepsilon) = x_1 - a_1\varepsilon, \quad x'(p_1 - \varepsilon) = a_1, \quad x(p_1 + \varepsilon) = x_1 + a_2\varepsilon, \quad x'(p_1 + \varepsilon) = a_2,$$

By looking for a parabolic connection $y = P(t) = A(t - p_1)^2 + B(t - p_1) + C$ as a solution of DDE (4) on the interval $[p_1 - \varepsilon, p_1 + \varepsilon]$, starting at the point $(p_1 - \varepsilon, x_1 - a_1\varepsilon)$ and satisfying the necessary initial-boundary conditions,

$$x(p_1 - \varepsilon) = x_1 - a_1\varepsilon, \quad x'(p_1 - \varepsilon) = a_1, \quad x'(p_1 + \varepsilon) = a_2,$$

one easily find that the values of the parameters $A, B$, and $C$ are

$$A = \frac{a_2 - a_1}{4\varepsilon}, \quad B = \frac{a_1 + a_2}{2}, \quad C = \frac{a_2 - a_1}{4}\varepsilon + x_1.$$

Then, by checking the value of the parabolic segment of the solution $x(t) = P(t)$ at $t = p_1 + \varepsilon$, one verifies that

$$\begin{aligned}
P(p_1 + \varepsilon) &= A(t - p_1)^2 + B(t - p_1) + C|_{t=p_1+\varepsilon} \\
&= \frac{a_2 - a_1}{4\varepsilon}\varepsilon^2 + \frac{a_1 + a_2}{2}\varepsilon + \frac{a_2 - a_1}{4}\varepsilon + x_1 = x_1 + a_2\varepsilon = x_1 + a_2(t - p_1)|_{t=p_1+\varepsilon},
\end{aligned}$$

which is the same value as the piecewise affine solution at that point.

Thus, the solutions to Equation (4) with continuous $f = f_\delta$ and $a = A_\delta$ are obtained from the solutions when $f = f_0$ and $a = A_0$ by simply cutting out and replacing the small corner segments with respective parabolic segments in their $\varepsilon$-vicinity ($\varepsilon = c \cdot \delta$). An

immediate consequence of this fact is that the periodic solutions and their stability persist under such small $\delta$-perturbations of $f_0$ and $a_0$ in DDE (4).

Indeed, the easiest way to see this is to consider the new starting point for $t, t = \varepsilon$, with $\varphi(\varepsilon) = H \neq 0$. Then, the corresponding shift map $S^T$ using period $T$ is dynamically equivalent (conjugate) to the respective map constructed in the case of $f_0$ and $a_0$. But the latter is equivalent to a composition of respective maps $G, F, F_1, F_2$, and $F_0$ derived in Section 3. If one would like to stay with the initial value $H := \varphi(0) \neq 0$ in the case of continuous $f_\delta, a_\delta, \delta > 0$, then a small adjustment in the considerations should be made. Namely, a new $h := H \pm c\varepsilon$ should be introduced, where $c\varepsilon$ is the difference between the two solutions of the differential delay, Equation (4), in the two cases when $\delta = 0$ and small $\delta > 0$. Such a difference is shown in the graph in Figure 4b and in the graph in Figure 5b (in both graphs, it is positive). In general, such a difference is independent of the value $x_1$ (or equivalent values in other analogous corner situations). Therefore, all the resulting maps $G, F, F_0, F_1$, and $F_2$ derived in Section 3 will become finite compositions of the affine maps of the form $h \mapsto \tilde{F}$, where $\tilde{F}(h) = G_*(h + c_1\varepsilon) + c_2\varepsilon$, and $G_*(h)$ is one of the elementary maps considered in Sections 3.1 and 3.2. Therefore, we are in a position to state the following.

**Theorem 4.** *Suppose that the parametric values $a_1, a_2, p_1$, and $p_2$ are such that the assumptions of either one of Theorems 1 or 2 are satisfied. Then, there exists $\delta_0 > 0$ such that for every $0 \leq \delta \leq \delta_0$ the delay differential of Equation (4) with functions $f = f_\delta(x)$ and $a = A_\delta(t)$ has a $C^1$-smooth periodic solution with the same type of stability as in the respective Theorems 1 or 2. Such a solution converges in uniform metric as $\delta \to +0$ to the respective periodic solution with the corner type discontinuity for $x'$ (when $f = f_0, a = A_0$).*

Note that the smoothing of both functions $f_0$ and $a_0$ in a $\delta$-neighborhood of their discontinuity set can be created such that the resulting continuous nonlinearities $f_\delta$ and $a_\delta$ are continuously differentiable or even of the $C^\infty$ class, and Theorem 4 still remains valid. Such $C^1$-smoothness is required in some cases when, e.g., one would need to study the Floquet multipliers of the periodic solutions and consider the corresponding linearized equation along the periodic solutions.

Indeed, it is an elementary fact that there are many solutions for the following approximation problem. Find a smooth function $F(t)$ defined on an interval $[t_1, t_2]$ such that it satisfies the given boundary conditions: $F(t_1) = x_1, F'(t_1) = m_1$ and $F(t_2) = x_2, F'(t_1) = m_2$. One of the possible solutions can be suggested as a polynomial of degree 3 or higher. In our case of a $C^\infty$ smooth connection for $f_0(x)$ on the interval $[-\delta, \delta]$, one can use the following function:

$$f^0(x) = e^{\frac{\delta x}{x - \delta}} - 1, \text{ for } x \in [0, \delta], \quad \text{and} \quad f^0(x) = -f^0(x), \text{ for } x \in [-\delta, 0].$$

It is straightforward to verify that such a connecting function is decreasing on the interval $[-\delta, \delta]$, in which it is of class $C^\infty$, with all the matching derivatives at $x = 0, x = -\delta$, and $x = \delta$. An analogous $C^\infty$-connection can be used for the coefficient $a(t)$ at points $0, p_1$, and $T = p_1 + p_2$.

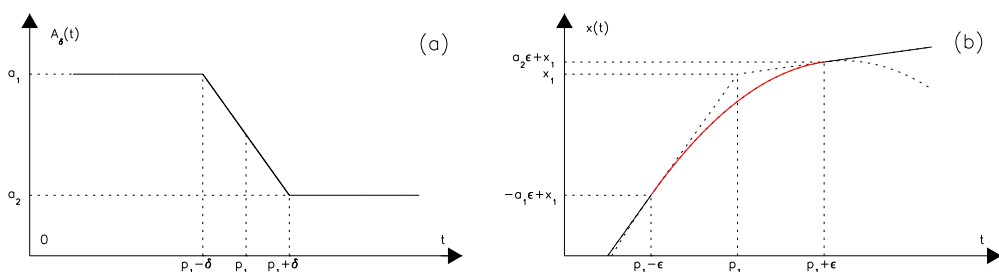

**Figure 5.** Sketches of (**a**) continuous coefficient $A_\delta(t)$ in $\delta$-vicinity of $t_1 = p_1$; (**b**) $C^1$-smooth solution with parabolic matching on $[p_1 - \varepsilon, p_1 + \varepsilon]$ for functions $A_\delta(t)$ and $f_\delta(x)$ with small $\delta > 0$.

**Author Contributions:** Conceptualization, A.I.; Methodology, A.I.; Software, S.S.; Validation, A.I. and S.S.; Formal analysis, A.I. and S.S.; Investigation, A.I. and S.S.; Writing – original draft, A.I. and S.S. All authors have read and agreed to the published version of the manuscript.

**Funding:** This research was financially supported by Simons Foundation (U.S.) through MATRIX-Simons Collaborative Fund and MATRIX-Simons Travel Grant Scheme, Flinders University (Australia) and Pennsylvania State University (U.S.).

**Data Availability Statement:** Data are contained within the article.

**Acknowledgments:** The authors thank the mathematical research institute MATRIX in Australia where part of this research was performed. This work's final version resulted from the collaborative activities of the authors during the workshop "Delay Differential Equations and Their Applications" (https://www.matrix-inst.org.au/events/delay-differential-equations-and-their-applications/ (accessed on 11 May 2024)) held in December 2023. The authors are also grateful for the financial support provided for these research activities by Simons Foundation (USA), Flinders University (Australia), and the Pennsylvania State University (USA). A.I.'s research was also supported in part by the Alexander von Humboldt Stiftung (Germany) during his visit to Justus-Liebig-Universität, Giessen, in June–August 2023.

**Conflicts of Interest:** The authors declare no conflict of interest.

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
