# Peer review of "Periodic Solutions in a Simple Delay Differential Equation"

_mca, doi:10.3390/mca29030036_

Round 1
Reviewer 1 Report
Comments and Suggestions for Authors
This paper examines a particular DDE with piecewise constant nonlinearities and a piecewise constant periodic coefficient. I struggled a bit to figure out how dependent the results were on assumed details. I think that the paper would be more interesting if the authors were able to comment on that. More specific comments below.
The authors don't give a clear rationale for studying equation (3). There are lots of systems of the form (2) in practical applications. It would be useful to readers if the authors could give one or more examples of the applicability of (3). Are there control systems modelled by equations of this form? If the authors don't know any systems of the form (3), is studying this system a stepping stone to studying more general equations? I think it's particularly important for the authors to comment on this because they go on to further simplify even this simple equation by making a(t) and f(x) piecewise constant. This is a nice way to obtain analytic results, but then there ought to be a good reason for studying the underlying nonlinear equation.
I think the authors need to be much more explicit about what they are assuming about p1 and p2. When these parameters are first introduced around equation (6), it looks like these are arbitrary positive constants. However, I believe from later comments (and from some of their proofs and computational work) that the authors are assuming that min(p1,p2) > 2. There may be other assumptions hiding in their work. Is there a good reason to limit values of p1 and p2 in this way from the point of view of applications? Is this some kind of assumption that the control system is (in some sense) reasonably fast?
Do the authors have any comments about what would happen if arbitrary values of p1 and p2 were allowed? In particular, what if p1 + p2 < 1? I don't think that the method of proof the authors used would extend to this case, but my instinct tells me that stable periodic solutions could still be obtained.
Continuing on the issue of p1 and p2 values considered in the numerical examples, I would have liked to have seen some situations where p1 + p2 was incommensurable with the delay value of 1. (I realize that this is not entirely possible in a digital computer. But one can certainly pick values whose least common multiple with 1 is a very large number.) Unless I missed something, the results presented should not depend on whether p1 and p2 are rational, but the range of numerical examples explored is limited to this case.
Minor comments:
Why do the section numbers start at zero? That's a bit strange.
In the second line of Proposition 1, I would use a strict inequality in 0 ≤ a(t). The next line introduces a minimum condition on a(t) that clearly excludes the equality.
In the last line on page 11, I think that "sate" should be "same".
Author Response
Please find the PDF document with reply attached.

Reviewer 2 Report
Comments and Suggestions for Authors
In this interesting work, the authors further investigate periodic solutions of an extension of the classical delay-differential equation representing feedback in, amongst others, physiological regulatory mechanisms. Their extension consists in making some of the coefficients time-periodic (instead of constant). This is part of a global research programm, and is presented as such by the authors, for which they obtain significant results in a special case (their eq. (3) being a special case of their eq. (2)).
The techniques of analysis are not really new, since the approximation of continuous feedback functions by piecewise constant functions is a by now a classical technique for these systems: a proof of convergence as the smoothness is reintroduced is also provided. The results presented are closely linked ("follow and expand on ideas and explicit constructions from" in the words of the authors) to those that have appeared in references [8] and [9], which are both Conference proceedings. The authors may want to more precisely distinguish the results presented there to those obtained here.
Author Response
Please find our reply to the referee attached as the PDF document.

Reviewer 3 Report
Comments and Suggestions for Authors
The research is presented in a state-of-the-art manner. I found no error. I recommend the article be ready for publication in its present form.
Author Response
We thank the reviewer for this positive comment.